# STD-FD: Spatio-Temporal Distribution Fitting Deviation for AIGC Forgery Identification

**Hengrui Lou** [1] **Zunlei Feng** [* 1 2] **Jinsong Geng** [2] **Erteng Liu** [1] **Jie Lei** [3] **Lechao Cheng** [4] **Jie Song** [1 2]
**Mingli Song** [1 5] **Yijun Bei** [* 1 2]

## Abstract

With the rise of AIGC technologies, particularly diffusion models, highly realistic fake images that can deceive human visual perception has become feasible. Consequently, various forgery detection methods have emerged. However, existing methods treat the generation process of fake images as either a black-box or an auxiliary tool, offering limited insights into its underlying mechanisms. In this paper, we propose Spatio-Temporal Distribution Fitting Deviation (STD-FD) for AIGC forgery detection, which explores the generative process in detail. By decomposing and reconstructing data within generative diffusion models, initial experiments reveal temporal distribution fitting deviations during the image reconstruction process. These deviations are captured through reconstruction noise maps for each spatial semantic unit, derived via a super-resolution algorithm. Critical discriminative patterns, termed DFactors, are identified through statistical modeling of these deviations. Extensive experiments show that STD-FD effectively captures distribution patterns in AIGC-generated data, demonstrating strong robustness and generalizability while outperforming state-of-the-art (SOTA) methods on major datasets. The source code is available at this link.

---

* Corresponding author. [1]State Key Laboratory of Blockchain and Data Security, Zhejiang University [2]School of Software Technology, Zhejiang University [3]College of Computer Science, Zhejiang University of Technology [4]School of Computer Science and Information Engineering, Hefei University of Technology [5]Hangzhou High-Tech Zone (Binjiang) Institute of Blockchain and Data Security. Correspondence to: Yijun Bei <beiyj@zju.edu.cn>.

*Proceedings of the 42$^{nd}$ International Conference on Machine Learning*, Vancouver, Canada. PMLR 267, 2025. Copyright 2025 by the author(s).

## 1. Introduction

With the rapid development of AIGC technologies (Cao et al., 2023), substantial advancements have been made in abstract concept and content generation. However, AIGC also presents significant risks. While it revolutionizes production methods and transforms industries such as media, entertainment, e-commerce, and education, it also introduces major security concerns, particularly in image forgery. Previous studies (Zhao et al., 2023) have highlighted that malicious actors can exploit AIGC to forge and manipulate images, making it increasingly difficult to verify the authenticity of generated content. This undermines trust in multimedia information and poses serious societal threats, including financial fraud, misinformation, and identity theft.

The rapid advancement of AIGC generation technologies has made effective forgery detection of new types of AIGC-generated images an urgent challenge. Unlike traditional forgery methods, which involve region manipulation, editing, or recreation, AIGC employs novel generative frameworks, such as diffusion models. This shift renders conventional forgery detection techniques, which are primarily designed for GAN-based tasks, ineffective for identifying forgery in diffusion-generated images. In response to these challenges, recent studies have focused on designing deepfake detection models specifically targeting diffusion-generated images. These approaches (Sha et al., 2023; Corvi et al., 2023b; Wang et al., 2023; Chen et al., 2024) primarily rely on reconstruction errors between real and fake images, treating reconstruction as an end-to-end tool for identification. However, this strategy is highly sensitive to the reconstruction model itself, with its properties significantly influencing forgery detection performance. For instance, a reconstruction model pre-trained on animal images may produce larger errors when applied to plant images, misclassifying real plant images as fake. Furthermore, reconstruction-based methods often treat the diffusion model's generative process as a black-box, limiting their ability to capture the deeper mechanisms underlying fake image generation. By focusing solely on reconstruction errors, these methods fail to fully analyze the generative process, reducing their effectiveness in detecting modern AIGC content.

To address these limitations, we propose a novel approach based on analyzing the distribution fitting of forged data through the lens of generative diffusion models, which involves stepwise decomposition and reconstruction. Specifically, to effectively capture the differences in distribution fitting across different semantic modules, we first employ a superpixel algorithm to partition the image into minimal units based on color and positional features from a 2D spatial perspective. Next, a generative diffusion model is adopted to obtain the distribution of stepwise decomposition and reconstruction for each semantic unit. We then model the distribution fitting deviation for each semantic unit to extract fitting differences. Finally, these distribution fitting deviations are aggregated to construct forgery identification features, which achieve state-of-the-art performance using a simple classifier. Our key contributions are as follows:

- We present a spatio-temporal distribution fitting framework that extracts temporal change patterns of each semantic block for AIGC forgery identification, offering a novel perspective on distribution fitting deviations for future deepfake research.

- Building on the underlying decomposition and reconstruction mechanisms of the diffusion process, discriminant change patterns DFactor is devised to extract fitting deviations of spatio-temporal distribution for AIGC forgery identification.

- Extensive experiments demonstrate that the proposed STD-FD achieves state-of-the-art performance in forgery identification task and significantly improves generalization capability in real-world scenarios.

## 2. Related Works

### 2.1. Forgery Technology

GAN-based and autoregressive models were pioneers in the field of image generation, with notable products such as BigGAN (Brock et al., 2018) and DALL·E (Open AI, 2023) emerging during their development. However, limitations in their control over generated content, along with stability and complexity issues, paved the way for a new image generation paradigm: diffusion models. Denoising Diffusion Probabilistic Models (DDPM) (Ho et al., 2020) demonstrated promising generative quality, sparking a series of studies on diffusion models. Subsequent research focused on enhancing structural design and improving sampling efficiency. Later, the Latent Diffusion Model (LDM) (Rombach et al., 2021) applied the diffusion process in latent space, significantly improving diffusion model efficiency and introducing text-conditioning capabilities through cross-attention mechanisms. LDM has since become a key driver in advancing image generation technology, leading to the development of prominent models such as Stable Diffusion (SD) (Stability.ai, 2023) and Midjourney (Midjourney, 2022).

### 2.2. Image Forgery Identification

Initially, researchers focused on identifying clues for detecting forged images in the spatial domain, such as color (McCloskey & Albright, 2018), saturation (McCloskey & Albright, 2019), and blended boundaries (Li et al., 2020). However, as image generation technologies advanced, it became increasingly difficult to construct reliable hand-crafted features in the spatial domain. Additionally, images often undergo multiple rounds of compression during transmission across various streaming platforms, resulting in low-quality images that obscure forgery artifacts. To address these challenges, researchers shifted towards exploring the frequency domain for discriminative clues, leading to methods based on different frequency bands (Li et al., 2021), scales (Wang et al., 2022), and adaptive frequency feature extraction (Qian et al., 2020a). While frequency-domain methods demonstrate strong forgery detection performance in highly compressed images, their effectiveness significantly decreases when confronted with unknown forgery techniques.

Most of the aforementioned works were designed for GAN-generated images. Although researchers (Corvi et al., 2023a; Ricker et al., 2022) identified that spectral artifacts can arise in diffusion-generated images, their effectiveness in detecting fake images remains limited. To address this, researchers are increasingly moving away from traditional clues used for GAN-generated images and are seeking unique indicators derived from the diffusion generation process itself. For instance, DIRE (Wang et al., 2023) and SeDIE (Ma et al., 2023) leveraged reconstruction errors from diffusion models to detect fake diffusion-generated images. Building on this, LaRE2 (Luo et al., 2024) and AEROBLADE (Ricker et al., 2024) focused on reconstruction errors in the latent space for detection. DRCT (Chen et al., 2024) employed reconstruction results from both real and fake images, training a classifier using contrastive learning loss.

However, these methods primarily rely on reconstruction errors, which makes them highly susceptible to the influence of the pre-trained reconstruction model. If the training domain of the reconstruction model differs from the detection domain (e.g., training on cat images but applying it to detect forged images of dogs or even houses), real images may be misclassified as fake. The root cause of this issue lies in treating reconstruction results as mere metrics and using the diffusion reconstruction process only as an auxiliary tool, without fully analyzing the underlying diffusion mechanisms. This oversight prevents a comprehensive understanding of the generative principles behind fake data.

# 3. Preliminaries

In this section, we first define the key notations of the diffusion model and outline its fundamental principles. Building on this foundation, we then introduce the entry points of the proposed STD-FD method, which delves into the underlying mechanisms of the diffusion process.

## 3.1. Notations

In this section, we utilize the standard notations defined by DDPM (Ho et al., 2020). The true data distribution is represented as $q(x_0)$, while the latent variable model approximates it as $p_\theta(x_0)$. The noise-prediction model, $\epsilon_\theta$, is parameterized by weights $\theta$.

The diffusion model comprises a $T$-step diffusion process $q(x_t|x_{t-1})$ and a denoising process $p_\theta(x_{t-1}|x_t)$ for $1 \leq t \leq T$, which is defined as follows:

$$q(x_t|x_{t-1}) = \mathcal{N}(x_t; \sqrt{1-\beta_t}x_{t-1}, \beta_t\mathbf{I}), \quad (1)$$

$$p_\theta(x_{t-1}|x_t) = \mathcal{N}(x_{t-1}; \mu_\theta(x_t,t), \Sigma_\theta(x_t,t)), \quad (2)$$

where $x_t$ refers to the diffusion result at timestep $t$, $\beta_t$ is the noise factor at timestep $t$, $\mathbf{I}$ is the identity matrix, $\mathcal{N}$ represents the normal distribution, indicating that the data probabilities during both noise addition and denoising follow a normal distribution. $\mu_\theta$ and $\Sigma_\theta$ are the mean and variance matrix of the denoising distribution respectively. The forward sampling at time step $t$ is given as follows:

$$q(x_t|x_0) = \mathcal{N}(x_t; \sqrt{\bar{\alpha}_t}x_0, (1-\bar{\alpha}_t)\mathbf{I}), \quad (3)$$

where $\alpha_t = 1 - \beta_t$ and $\bar{\alpha}_t = \prod_{s=1}^{t} \alpha_s$. $\alpha_s$ denotes the noise weighting coefficient at each diffusion timestep, controlling the intensity of the noise added at each step.

## 3.2. Definitions

In this section, we follow the sampling process from DDIM (Song et al., 2021) and explore the key aspects based on distribution variations starting from the deterministic sampling function itself.

The deterministic sampling function recovers original data from noised input $x_0$ at timestep $t$, which is defined as:

$$x_t = \sqrt{\alpha_t}x_0 + \sqrt{1-\alpha_t}\epsilon_\theta^{(t)}(x_t), \quad (4)$$

where $\epsilon_\theta^{(t)}(x_t)$ is the noise predicted by the network at timestep $t$.

In combination with the following equation:

$$x_0 = \frac{x_t - \sqrt{1-\alpha_t}\epsilon_\theta^{(t)}(x_t)}{\sqrt{\alpha_t}}, \quad (5)$$

$$x_{t-1} = \sqrt{\alpha_{t-1}}x_0 + \sqrt{\frac{1-\alpha_{t-1}-\sigma_t^2}{1-\alpha_t}}(x_t - \sqrt{\alpha_t}x_0) + \sigma_t^2\epsilon_t, \quad (6)$$

The final sampling formula (corresponding to Equation (12) in the DDIM paper) is obtained as follows:

$$x_{t-1} = \sqrt{\alpha_{t-1}} \underbrace{\left( \frac{x_t - \sqrt{1-\alpha_t}\epsilon_\theta^{(t)}(x_t)}{\sqrt{\alpha_t}} \right)}_{\text{"predicted } x_0\text{"}} \\ + \underbrace{\sqrt{1-\alpha_{t-1}-\sigma_t^2} \cdot \epsilon_\theta^{(t)}(x_t)}_{\text{"direction pointing to } x_t\text{"}}. \quad (7)$$

where $\sigma_t$ is set to 0 to achieve a deterministic sampling process, and thus the random noise term $\sigma_t^2\epsilon_t$ is removed.

From Eq. (7), we observe that for any timestep $x_{t-1}$ during the sampling process, $\alpha_{t-1}$ is the known noise factor determined by the scheduling scheme. The unknown parameters are "predicted $x_0$" and "direction pointing to $x_t$". Further, both parameters are derived from the same variable $\epsilon_\theta^{(t)}(x_t)$. Therefore, the changes in the data and distribution during the diffusion model's sampling process are driven by noise $\epsilon_\theta^{(t)}(x_t)$. As a result, STD-FD constructs the key representations of distributional variations by capturing distributed noise $\epsilon_\theta^{(t)}(x_t)$ at each timestep of the sampling process.

# 4. Methodology

In this section, we first outline the extraction of spatio-temporal distribution change patterns and the construction of global discriminative factors. Next, the discrepancy detection module is devised to identify spatio-temporal distribution fitting deviation for forged image identification.

## 4.1. Spatio-Temporal Distribution Change Extraction

**Noise Reshaping.** As mentioned in the preliminaries, we construct key representations that capture the changes throughout the entire diffusion sampling process by tracking distributed noise $\epsilon_\theta^{(t)}(x_t)$ at each timestep. $\epsilon_\theta^{(t)}(x_t)$ represents the noise added to $x_0$ at each timestep, with a shape of $(C, H, W)$. To facilitate the subsequent capture of spatial and temporal information during the sampling process, we convert $\epsilon_\theta^{(t)}(x_t)$ to image format, as shown in the following:

$$Norm(\epsilon_\theta^{(t)}(x_t)) = \frac{\epsilon_\theta^{(t)}(x_t) - \min(\epsilon_\theta^{(t)}(x_t))}{\max(\epsilon_\theta^{(t)}(x_t)) - \min(\epsilon_\theta^{(t)}(x_t))} \times 255, \quad (8)$$

where $Norm(\cdot)$ performs the normalization and converts the data into an image with values in the range $[0, 255]$.

**Spatial Information Capture.** To capture the spatial information of the noise itself during the sampling process,

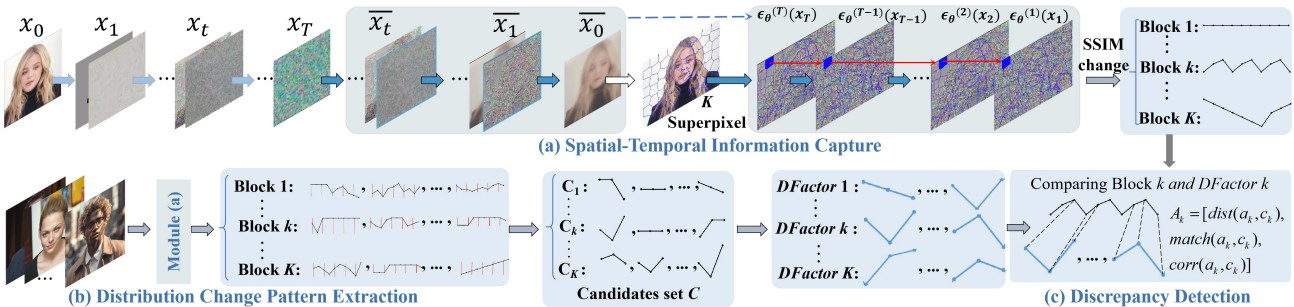

Figure 1. Framework of the proposed STD-FD, which is composed of two stages: spatial-temporal information capture (a & b) and discrepancy detection via distribution fitting deviation (c). In the first stage, temporal change information for each semantic block, corresponding to the spatial superpixel, is firstly extracted (**a**). Then, we focus on extracting the $DFactor$, which can distinguish between positive and negative samples during the sampling process. Next, global classification discriminative factors $DFactor$ are constructed by identifying key changing segments that differentiate between the two types of samples (**b**). Finally, based on the $DFactor$ extracted from the first stage, we perform distribution fitting deviation modeling on the data to be identified. Feature engineering is completed using distance, correlation, and matching metrics (**c**). With the extracted feature $A_k$, a classifier is trained for forgery identification.

we employ the SLIC (Achanta et al., 2012) superpixel to segment the image into minimal units based on color and spatial features from a two-dimensional perspective of the noise:

$$g(Norm(\epsilon_\theta^{(t)}(x_t))) = \{S_1, S_2, \ldots, S_K\}, \quad (9)$$

where $g(\cdot)$ denotes the application of the SLIC superpixel segmentation on the input image, and the image is divided into $K$ segments.

**Temporal Information Capture.** We capture the temporal information of $\epsilon_\theta^{(t)}(x_t)$ distribution changes from the diffusion sampling process. For a sampling process with $T$ timesteps, block $S_k$ contains intermediate sampling results over $T$ steps, i.e., $[S_k^1, S_k^2, \ldots, S_k^T]$. We record the temporal variation of the $\epsilon_\theta^{(t)}(x_t)$ by calculating the SSIM (Wang et al., 2004) between adjacent timesteps.

$$h_k = \{h_k^1, h_k^2, \ldots, h_k^{T-1}\}, h_k^t = \text{SSIM}(S_k^t, S_k^{t+1}), \quad (10)$$

where $h_k$ represents the temporal variation sequence of $\epsilon_\theta^{(t)}(x_t)$ for the $k$-th image block, with a sequence length of $T-1$, where $k \in \{1, 2, ..., K\}$. $\text{SSIM}(S_k^t, S_k^{t+1})$ denotes the Structural Similarity calculated between $S_k^t$ and $S_k^{t+1}$.

**Extraction of Distribution Change Patterns.** After capturing the spatial and temporal information, for an image to be analyzed, we can obtain $[h_1, h_2, \ldots, h_k, \ldots, h_K]$. Each $h_k$ represents the quantified time-series information of the noise distribution changes in the $k$-th block.

We now need to extract the distribution change patterns from each block. During the diffusion sampling process, there exists a temporal deviation in distribution modeling between real and fake samples. This deviation is manifested as

anomalies in certain time segments during the sampling process. Our current goal is to find such "anomalies" to serve as the basis for distinguishing positive and negative samples (Let's call it the Discriminant Factor, $DFactor$). Therefore, we have designed an algorithm to capture $DFactor$ from the time-series data that are crucial for classification.

Specifically, $H_k$ is the set of time-series information of noise distribution changes for the $k$-th block across all samples (including positive and negative samples), with the corresponding label set $Y_k$. We need to generate a candidate set of $DFactor$ $C_k$ (the specific steps for constructing the candidate set will be detailed in Section 4.2). For $C_k$, we first compute the weighted distance between each candidate $DFactor$ and the time series, then perform a weighted sum of all distances to obtain the total distance of the time series. We use GDTW (Greedy Dynamic Time Warping) to measure the distance; GDTW retains the nonlinear alignment characteristics of DTW while employing a greedy strategy to improve computational efficiency, making it suitable for handling long time-series data. The formula for calculating the weighted distance between each candidate $DFactor$ and the time series is as follows:

$$d(C_k, t) = \sqrt{\sum_{j=1}^{J} \left( \left( C_k^{\text{path}(j)} \cdot |w_{\text{local}}| \right) - t^{\text{path}(j)} \right)^2}, \quad (11)$$

where $J$ is the length of the alignment path obtained by the greedy algorithm. $C_k^{\text{path}(j)}$ and $t^{\text{path}(j)}$ are the $j$-th matched $DFactor$ and time-series segment according to the greedy alignment path, respectively. $w_{\text{local}}$ is the local timing factor within the $DFactor$, adjusting the importance of each position within the $DFactor$.

Next, we perform a weighted sum of all distances to obtain the total distance of the time series:

$$D(t) = \sum_{m=1}^{M} \text{Softmax} \left( -d \left( C_k^m, t \right) \cdot |w_{\text{global}}| \right) \cdot \\ \left( d \left( C_k^m, t \right) \cdot |w_{\text{global}}| \right), \quad (12)$$

where $M$ is the total number of candidates $C_k^m$. $w_{\text{global}}$ is the global factor, adjusting the importance of different $C_k^m$. Softmax assigns higher probabilities to larger values. By applying a negative sign, candidate factors closer to the current temporal data are given greater weight.

At this point, we have obtained the distance between each candidate $DFactor$ and the time series (including both positive and negative samples). We construct a loss function using KL divergence, and by minimizing the loss via gradient descent (maximizing the distance between positive and negative samples), we achieve the extraction of the optimal discriminative factors. The defined loss function is:

$$\mathcal{L} = - \sum_{n=1}^{N} \text{KL} \left( \mathcal{N}(\mu_+, \sigma_+), \mathcal{N}(\mu_-, \sigma_-) \right) + \chi, \quad (13)$$

$$\chi = \alpha \|\mathbf{w}_{\text{local}}\|_p + \beta \|\mathbf{w}_{\text{global}}\|_p, \quad (14)$$

where $\chi$ represents the sum of the local and global factors. $\mu_+, \sigma_+$ and $\mu_-, \sigma_-$ are the mean and standard deviation of distances between positive and negative classes. $\mathbf{w}_{\text{local}}$ and $\mathbf{w}_{\text{global}}$ are the local and global temporal factors, respectively. $\alpha$ and $\beta$ are regularization coefficients. $p$ is the norm degree.

### 4.2. Discrepancy Detection via Distribution Fitting Deviation

**Candidate Discriminative Factors Generation.** In Section 4.2, we obtained the discriminative factors $DFactor_k$ that can distinguish between positive and negative samples. We perform forgery identification based on the fitting deviation between the data to be tested and the discriminative factors. Specifically, we process the sample to be tested in the same way as in Section 4.1 to obtain $[h_1, h_2, \ldots, h_K]$.

First, for each time series $h_k$, we extract all subsequences of length $L$ to form the set $SCO_k$ and compute the initial distance score $D(sco_{i,k})$ of each subsequence relative to the mean subsequence $s\bar{c}o_k$. Then, we merge all subsequences and corresponding distance scores into a global set $SCO$.

In the greedy algorithm, we iteratively select the subsequence $sco^*$ with the highest distance score from $SCO$, add it to the candidate set $\mathcal{C}$, and update the distance scores $D(sco)$ of the remaining subsequences to encourage diversity among candidates. We repeat this process until we obtain $M$ candidate $DFactor_k$. The final set $\mathcal{C}$ contains candidate $DFactor_k$ for subsequent analysis.

**Identification Feature Construction.** After obtaining the candidate discriminative factors for each block of the sample to be tested, we construct features by comparing the candidate discriminative factors of each block with the best discriminative factors of the corresponding block. Specifically, for the candidate discriminative factors $\{c_k\}$ of the $k$-th block of the sample and the discriminative factors $\{DFactor_k\}$ of the $k$-th block, we construct features $\{\text{A}_k\}$ based on distance, matching degree, and correlation. Then, we concatenate the features $\{\text{A}_k\}$ from the $K$ blocks to form the forgery discrimination features of the data.

Specifically, for each block $k$ ($k = 1, 2, \ldots, K$), we have the candidate discriminative factor $c_k$ of the sample, and the discriminative factor $DFactor_k$ obtained from training. Based on the distance, matching degree, and correlation between $c_k$ and $DFactor_k$, we compute the feature vector $\text{A}_k$ as follows:

$$\text{A}_k = [dist(c_k, a_k),\ match(c_k, a_k),\ corr(c_k, a_k)], \quad (15)$$

Where $dist(c_k, a_k)$, $match(c_k, a_k)$, and $corr(c_k, a_k)$ represent the distance, matching degree, and correlation between $c_k$ and $a_k$, respectively.

Then, we concatenate the feature vectors $\text{A}_k$ of all $K$ blocks to form the forgery discrimination feature vector A:

$$\text{A} = [\text{A}_1,\ \text{A}_2,\ \ldots,\ \text{A}_K]. \quad (16)$$

Using the above feature vector A, a forgery identification classifier is trained. During the testing phase, the feature vector of the test image is input into the classifier for forgery identification.

## 5. Experiments

In this section, we present a detailed evaluation of the performance of STD-FD across three parts. Specifically, we first provide comprehensive experimental settings. Next, we compare the performance of STD-FD with detection methods, including SOTA approaches, on multiple datasets. Then, we conduct experimental analyses of the generalization performance of STD-FD.

### 5.1. Experimental Setup

**Datasets and Evaluation Metrics.** We evaluate the proposed method using the GenImage (Zhu et al., 2023) and Deepfacegen (Bei et al., 2024) datasets. GenImage comprises 1,331,167 real images and 1,350,000 generated images. The generated images are sourced from 8 different generative model. In our research, we follow the official dataset split, allocating 2,581,167 images for training and reserving the remaining 100,000 images for validation. Deep-FaceGen is a facial forgery dataset that includes both video

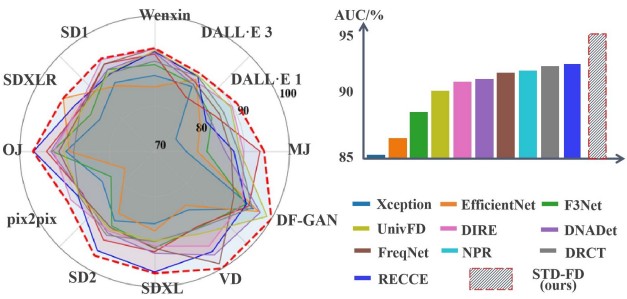

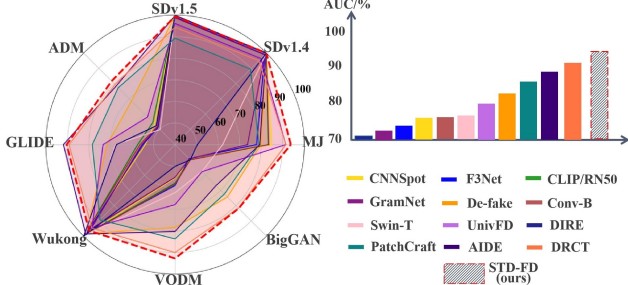

*Figure 2.* Performance comparisons on Deepfacegen (AUC, %). The radar chart on the left illustrates the detection performance of STD-FD compared to other methods, with solid lines representing the comparison methods and a dashed line representing STD-FD. The bar chart on the right presents the average AUC across 12 subsets for each method, where solid-edged bars indicate the comparison methods, and dashed-edged bars represent STD-FD.

*Figure 3.* Performance comparisons on Genimage (ACC, %). The radar chart on the left illustrates the detection performance of STD-FD compared to 12 other methods, with solid lines representing the comparison methods and a dashed line representing STD-FD. The bar chart on the right presents the average ACC across 8 subsets for each method, where solid-edged bars indicate the comparison methods, and dashed-edged bars represent STD-FD.

and image modalities. We selected facial forgery data generated using novel prompt-guided forgery methods. These methods encompass a total of 12 approaches based on both diffusion and autoregressive models, using either text-to-image or image-to-image modalities. Consistent with the evaluation metrics established in the benchmark paper, we adopt Accuracy (ACC) and Area Under the Curve (AUC) as our performance measures.

**Implementation Details.** XGBoost (Chen & Guestrin, 2016) is employed as the classifier. In the identification feature construction phase, Euclidean distance and Dynamic Time Warping (DTW) are adopted to measure the similarity between the candidate discriminative factors and the optimal discriminative factors. For match rate-based methods, trend match rate (using sign matching) is applied, while Pearson correlation and template matching (NCC) are employed for correlation-based methods. The experiments are conducted on two GeForce RTX 4090 GPU (24GB VRAM).

### 5.2. Comparison with SOTAs

**Deepfacegen Benchmark.** In this section, we compare our method with several mainstream and state-of-the-art generated image detection methods, including Xception (Chollet, 2017), EfficientNet-B0 (Tan & Le, 2020), F3-Net (Qian et al., 2020b), RECCE (Cao et al., 2022), DNADet (Yang et al., 2022), DIRE (Wang et al., 2023), DRCT (Chen et al., 2024), UnivFD (Ojha et al., 2023), NPR (Tan et al., 2024a), and FreqNet (Tan et al., 2024b). Following the evaluation protocol of DeepFaceGen, our proposed method, STD-FD, was trained and tested in the same manner as the comparison methods on both real data and data generated by 12 different forgery methods, including DALL·E (Open AI, 2023) and SD (Stable Diffusion) (Stability.ai, 2023). The dataset was split into training, validation, and test sets in a ratio of 7:1:2. The results, shown in Figure 2, indicate that STD-FD achieved the best performance across all 12

subsets. Notably, the improvement was most significant in the subsets where distinguishing between real and fake images was particularly challenging, such as those generated by DALL·E 1, DALL·E 3, Midjourney, and Wenxin.

Overall, STD-FD achieved AUC exceeding 90% across all subcategories of forged data, with an average AUC of 94.90% (detailed results are in Table 9 in the Appendix).

**Genimage Benchmark.** To further validate the effectiveness of STD-FD, we conducted comparisons following the same experimental protocol as GenImage. We compared our method with 12 forgery detection models, including CNNSpot (Wang et al., 2020), F3Net (Qian et al., 2020b), CLIP/RN50 (Radford et al., 2021), GramNet (Liu et al., 2020), De-fake (Sha et al., 2023), Conv-B (Liu et al., 2022), Swin-T (Liu et al., 2021), UnivFD (Ojha et al., 2023), DIRE (Wang et al., 2023), PathCraft (Zhong et al., 2023), AIDE (Yan et al., 2024), and DRCT (Chen et al., 2024). All identification methods were trained on the SDv1.4 subset of GenImage. Specifically, STD-FD and the comparison methods were trained on SDv1.4 and evaluated on different testing subsets. This benchmark poses a significant challenge, as the overall accuracy is closely tied to the cross-generator generalization ability of the identification methods. The comparative results in Figure 3 show that all methods achieve very high identification accuracies on the SDv1.4, SDv1.5, and Wukong subsets. However, a noticeable drop in accuracy (ACC) can be observed across other subsets such as Midjourney, ADM, GLIDE, VQDM, and BigGAN, particularly for non-diffusion-based generators like BigGAN. In contrast, STD-FD demonstrated superior performance on these challenging subsets. Even on BigGAN, which is based on GAN architectures, STD-FD achieved the best results. This can be attributed to the DFactor extraction process of STD-FD, which spans both the temporal and spatial domains, enabling excellent generalization performance by distinguishing between positive and negative samples.

*Table 1.* Generalization Performance Comparison on Genimage Subsets with Mainstream and SOTA Methods. Each value represents (Score1 / Score2), where Score1 denotes the ACC trained and tested on the current subset, and Score2 denotes the average ACC trained on the current subset and tested on other subsets.

| Method | Midjourney | SDv1.4 | SDv1.5 | ADM | GLIDE | Wukong | VQDM | BigGAN |
|---|---|---|---|---|---|---|---|---|
| DIRE | 99.5/72.6 | 99.7/68.8 | 99.7/68.8 | 99.2/67.9 | 99.8/52.2 | 99.7/68.1 | 99.9/51.9 | 100/50.1 |
| Defake | 98.8/78.7 | 99.8/81.5 | 100/82.3 | 98.7/82.5 | 99.8/73.1 | 99.3/75.6 | 99.1/82.0 | 98.7/76.6 |
| PatchCraft | 90.1/81.1 | 89.5/81.3 | 90.1/81.3 | 90.4/79.3 | 90.6/80.4 | 92.6/79.3 | 90.3/74.9 | 80.9/73.7 |
| AIDE | 98.9/86.0 | 99.7/84.9 | 99.6/84.0 | 95.3/79.1 | 97.6/81.8 | 98.8/80.4 | 92.3/83.0 | 96.3/81.7 |
| DRCT | 99.9/89.3 | 95.5/86.6 | 96.8/89.1 | 95.1/81.8 | 96.6/83.8 | 99.6/84.4 | 94.3/82.4 | 98.8/84.0 |
| **STD-FD** | **99.9/91.9** | **99.9/90.8** | **100/89.8** | **99.5/84.7** | **100/85.9** | **100/86.3** | **99.7/84.4** | **100/86.9** |

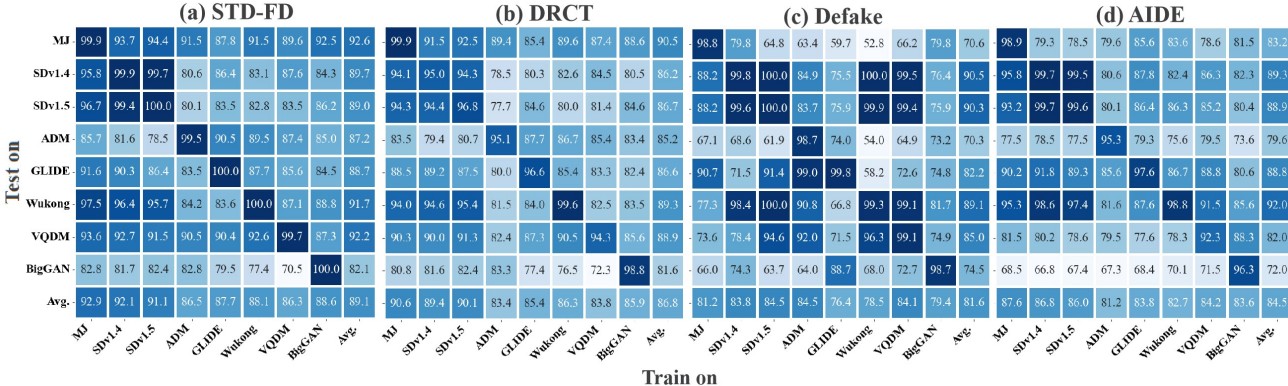

*Figure 4.* Results (ACC, % ) of cross-validation on different training and testing subsets. For each generator, we train a model and test it on all 8 generators. (a) illustrates the generalization performance of our proposed method, STD-FD, across different subsets of the GenImage dataset, while (b-d) present the generalization results of the sota models.

In summary, STD-FD achieves the highest ACC in five out of eight identification subsets, and its average ACC surpasses that of existing forgery identification methods. This highlights the consistent effectiveness of STD-FD (detailed results are provided in Table 10 in the Appendix).

### 5.3. Generalization Performance

To validate the generalizability of the STD-FD, we designed two experimental scenarios: one focusing on constructing data from previously unseen fake forgery methods (i.e., methods not present in the model's training data), and the other examining cases where the pre-trained reconstruction model is mismatched with the target recognition task (e.g., as previously mentioned, training a model on cat images but applying it to detect forged dog or even house images).

**Addressing Unknown Forgery Methods.** Following the generalization performance evaluation setup in (Luo et al., 2024), we evaluate the performance of our model on the GenImage dataset by training on one subset and testing on all eight subsets. The experimental results of STD-FD are compared with the baseline model DIRE and the top four models currently ranked on GenImage. As shown in Table 1, STD-FD achieves the best generalization performance across all eight subsets compared to the competing

*Table 2.* Results (AUC, % ) of Mismatch Between Pre-trained Model and Identification Target. The evaluation follows Deepfacegen Benchmark , identification methods are trained and tested on deepfacegen consisting of real data and forged data generated by 12 prompt-guided methods.

| Method \ Pre-trained Dataset | Cat | Horse | Bedroom |
|---|---|---|---|
| DIRE (Wang et al., 2023) | 73.01 | 65.45 | 55.04 |
| DRCT (Chen et al., 2024) | 74.48 | 68.78 | 67.62 |
| STD-FD (ours) | 92.78 | 92.45 | 92.77 |

algorithms. Moreover, the detailed comparison with the current SOTA models, DRCT (Fig 4 (b)), demonstrates that identification metrics inevitably decline when confronted with unknown forgery methods different from the training data. However, compared to the SOTA method on GenImage, DRCT, STD-FD maintains superior generalization ability. This can be attributed to STD-FD's comprehensive extraction of classification features that distinguish real and fake data across both temporal and spatial domains. The approach of modeling the distribution fitting deviation between real and fake data exhibits excellent generalization.

**Mismatch Between Pre-trained Model and Identification Target.** As mentioned in the Introduction, existing

*Table 3.* Identification performance (ACC,%) when choosing different classifiers. STD-FD with different classifiers are trained on SDv1.4 and evaluated on different testing subsets.

| Classifier | Midjourney | SDv1.4 | SDv1.5 | ADM | GLIDE | Wukong | VQDM | BigGAN | Avg. |
|---|---|---|---|---|---|---|---|---|---|
| LR | 91.41 | **99.99** | **99.98** | 80.07 | 88.84 | 95.82 | 89.76 | 79.84 | 90.71 |
| RF | 93.37 | 99.98 | 98.24 | 81.18 | 87.79 | 95.29 | 91.68 | 81.03 | 91.07 |
| SVM | 92.45 | 99.98 | 99.01 | 81.57 | 89.80 | 95.43 | 91.55 | 80.88 | 91.33 |
| Xgb | **93.76** | **99.99** | 99.45 | **81.64** | **90.31** | **96.41** | **92.74** | **81.79** | **92.01** |

forgery detection methods based on reconstruction errors exhibit a catastrophic drop in performance when the data type for forgery detection is inconsistent with the pre-trained reconstruction model. For example, reconstruction models pre-trained on animals may exhibit larger reconstruction errors when applied to plants, causing real plant images to be misclassified as fake. Therefore, we conduct generalization experiments under such scenarios for STD-FD. Specifically, we use the LSUN bedroom, LSUN cat, and LSUN horse pre-trained reconstruction models provided by (Dhariwal & Nichol, 2024), and compare the generalization performance with two reconstruction error-based detection methods, DIRE and DRCT. The evaluation follows Deepfacegen Benchmark, involving training and testing on 12 prompt-guided forgery datasets. The experimental results, as shown in Table 2, reveal a catastrophic drop in generalization performance for the reconstruction error-based methods, while STD-FD maintains high generalization capability. Notably, the AUC scores for STD-FD remain above 92% under the cat, horse, and bedroom pre-training scenarios, demonstrating that STD-FD is not significantly affected by the training scenarios of the pre-trained reconstruction models.

## 6. Ablation Study

**Influence of Classifier.** We evaluated STD-FD with classifiers beyond XGBoost (XGB), including Logistic Regression (LR), Random Forest (RF), and Support Vector Machine (SVM), using the Genimage Benchmark setup. Results in Table 3 show that classifier choice has minimal impact, with the average AUC exceeding 90% across all classifiers, further confirming STD-FD's robustness.

**Influence of Sampling Step.** To evaluate the effect of sampling timesteps on STD-FD, we conduct experiments on four challenging subsets of DeepFaceGen with sampling timesteps $T$ set to 5, 10, 20, and 50. Figure 5 show that the AUC scores of STD-FD consistently exceed 90% across all timestep settings, confirming the method's robustness.

**Influence of Post-Processing.** To assess the resilience of STD-FD against post-processing, we follow the experimental setup from (Wu et al., 2023), applying resizing and JPEG compression to images. Using the same setup as in the "Influence of Sampling Step" part, the results in Table 4 show that STD-FD consistently achieves AUC scores above 90%,

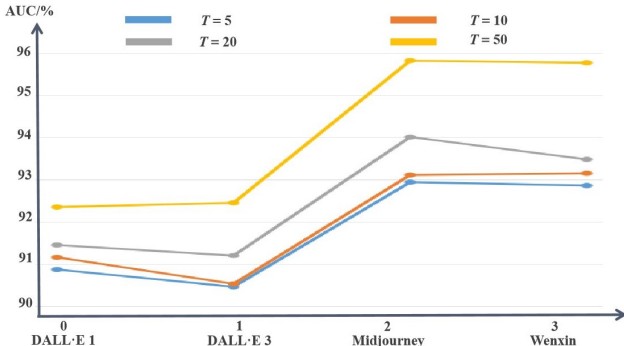

*Figure 5.* Performance (AUC,%) on four challenging subsets of DeepFaceGen when choosing different sampling timestep $T$.

*Table 4.* Performance (AUC,%) when choosing different post-processings on four challenging subsets of DeepFaceGen.

| Post-Processing | | Dataset Subset | | | |
|---|---|---|---|---|---|
| Scale | QF | DALL·E 1 | DALL·E 3 | Midjourney | Wenxin |
| 0.50 | - | 91.45 | 91.25 | 94.01 | 93.22 |
| 0.75 | - | 91.51 | 91.20 | 94.01 | 93.51 |
| 1.00 | - | 91.45 | 91.20 | 94.01 | 93.48 |
| 1.25 | - | 91.57 | 91.20 | 93.78 | **94.01** |
| 1.50 | - | **91.62** | 91.07 | **94.11** | 93.78 |
| - | 60 | 90.96 | 90.97 | 93.87 | 93.11 |
| - | 70 | 90.87 | 91.04 | 93.87 | 93.45 |
| - | 80 | 91.03 | 91.43 | 93.87 | 93.21 |
| - | 90 | 91.45 | **91.54** | 93.95 | 93.07 |
| - | 100 | 91.45 | 91.20 | 94.01 | 93.48 |

with fluctuations under 1%, demonstrating its robustness against various post-processing strategies.

**Influence of Spatial and Temporal Modules.** Given the inherent temporal structure of the STD-FD network, we focus on ablating the spatial module. In the DeepFaceGen benchmark setup, when superpixel segmentation is omitted, the AUC drops from 94.90% to 91.12%, but performance remains superior to other state-of-the-art methods. This underscores the importance of the spatial module and demonstrates the effectiveness of the temporal module's feature extraction in providing a robust engineering solution.

**Influence of Adversarial Attacks.** We selected FGSM (Goodfellow et al., 2015), PGD (Madry et al., 2017), C&W (Carlini & Wagner, 2016), and Black-Box attacks (Guo et al., 2019), based on STD-FD's feature mod-

*Table 5.* Performance (AUC,%) when facing different attacks. on four challenging subsets of DeepFaceGen.

| Attack Type | DALL·E 1 | DALL·E 3 | Midjourney | Wenxin | Average |
|---|---|---|---|---|---|
| Raw | 91.45 | 91.20 | 94.01 | 93.48 | 92.53 |
| FGSM | 85.01 | 86.21 | 88.64 | 90.21 | 87.51 |
| PGD | 84.21 | 85.01 | 84.65 | 87.02 | 85.22 |
| C&W | 83.89 | 84.45 | 85.63 | 88.32 | 85.57 |
| Black-Box | 86.64 | 87.32 | 90.22 | 89.56 | 88.43 |

*Table 7.* Performance (AUC,%) when facing different K blocks on four challenging subsets of DeepFaceGen.

| $K$ | DALL·E 1 | DALL·E 3 | Midjourney | Wenxin | Performance |
|---|---|---|---|---|---|
| 1 | 90.46 | 90.11 | 92.89 | 92.67 | -1.08% |
| 5 | 91.45 | 90.87 | 93.76 | 94.01 | -0.01% |
| 10 | 91.45 | 91.20 | 94.01 | 93.48 | Baseline |
| 15 | 91.89 | 91.01 | 93.45 | 92.93 | -0.23% |
| 20 | 92.45 | 90.89 | 93.87 | 91.54 | -0.37% |

eling and classifier attributes. Using data-based adversarial attacks from the preliminary DeepFaceGen experiments, we observed that, although STD-FD's performance decreased from the original 93.48%, the worst performance (85.22% under PGD) still showed a drop of less than 7%, maintaining a high overall AUC.

**Influence of Adversarial Perturbations.** Following the same experimental settings described in the Influence of Adversarial Attacks section, we conducted additional experiments specifically focusing on adversarial perturbations during the diffusion sampling process. Concretely, adversarial noise (with L2-norm strengths of [0.01, 0.03, 0.05]) was injected at each timestep of the reverse diffusion process (20 steps in total). Under these adversarial conditions, the performance fluctuates within approximately 2.5%, demonstrating that the STD-FD identification mechanism remains robust against targeted sampling attacks.

*Table 6.* Performance (AUC,%) when facing adversarial perturbations on four challenging subsets of DeepFaceGen.

| Perturbation | DALL·E 1 | DALL·E 3 | Midjourney | Wenxin | Performance |
|---|---|---|---|---|---|
| 0.05 | 88.01 | 89.98 | 93.84 | 92.57 | -2.5% |
| 0.03 | 89.56 | 90.43 | 93.56 | 93.62 | -1.8% |
| 0.01 | 91.67 | 91.42 | 92.90 | 90.42 | -2.0% |
| Original | 91.45 | 91.20 | 94.01 | 93.48 | Baseline |

**Influence of Block Parameters $K$.** We performed an ablation study varying the number of superpixel blocks K from the baseline setting K=10. The performance variation across different K values is within approximately 1.08%. It's noteworthy that selecting a larger value of K does not always yield better results. Superpixel methods inherently suggest an optimal clustering number based on image content. In facial forgery scenario, the recommended K ensures effective semantic consistency; significantly deviating from this value negatively affects pixel-level semantic coherence and impairs spatio-temporal decoupling during diffusion sampling.

## 7. Conclusion

In this paper, we introduce STD-FD, a novel approach that uncovers the generative mechanisms behind AIGC-generated fake images, particularly those produced by diffusion models. STD-FD systematically explores the tem-

poral distribution deviations during image reconstruction. By leveraging superpixel-based semantic segmentation, we model these deviations as spatio-temporal $DFactor$ across different semantic modules. Extensive experiments demonstrate the effectiveness of STD-FD, proving its robustness and generalizability. Future work will focus on developing an end-to-end framework to uncover additional distinctive spatio-temporal patterns for AIGC forgery detection.

## Acknowledgements

This work is funded by National Key Research and Development Project (Grant No: 2022YFB2703100) and Information Technology Center, Zhejiang University.

## Impact Statement

This paper presents work aimed at advancing deepfake detection. Our research contributes to improving the reliability and security of AI-generated content detection, helping mitigate potential risks associated with deepfake technology. While there are various societal implications of our work, none which we feel must be specifically highlighted here.

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

# STD-FD: Spatio-Temporal Distribution Fitting Deviation for AIGC Forgery Identification

In the appendix, we first present the detailed pseudocode of STD-FD (A) to facilitate a better understanding of the method. To further illustrate the key issues addressed by STD-FD and its design philosophy, we provide an example analysis (B). We then analyze the actual runtime and resource usage of STD-FD (C). Finally, the Detailed Experimental Results section presents the comprehensive results of the main benchmark experiments (D).

## A. STD-FD Pseudo-code

To further clarify the implementation process of STD-FD, we provide the core code for its key steps in addition to the open-source code. Specifically, algorithms 1 and 2 detail the generation and selection process of the critical spatio-temporal modeling factor, $DFactor$, in STD-FD.

---

**Algorithm 1** Learning Discriminative Factor

**Input:** $H_k, Y_k, C_k$
**Output:** Top $N$ discriminative factors
1: Initialize empty list $DFactors$
2: **for** each candidate $c$ in $C_k$ **do**
3:     Initialize timing factors $w_{\text{local}}, w_{\text{global}}$
4:     **for** $iter = 1$ to $MaxIterations$ **do**
5:         i. Compute distances:
6:         $Distances \quad = \\ \sum\limits_{m=1}^{M} \text{Softmax} \left( -d\left(C_k^m, t\right) \cdot |w_{\text{global}}| \right) \\ \times \left( d\left(C_k^m, t\right) \cdot |w_{\text{global}}| \right)$
7:         ii. Compute loss:
8:         $Loss = -\sum\limits_{n=1}^{N} \text{KL}\left( \mathcal{N}(\mu_+, \sigma_+), \mathcal{N}(\mu_-, \sigma_-) \right) \\ + \alpha \|\mathbf{w}_{\text{local}}\|_p + \beta \|\mathbf{w}_{\text{global}}\|_p$
9:         iii. Update timing factors: $(w_{\text{local}}, w_{\text{global}})$
10:     **end for**
11:     Add $(c, w_{\text{local}}, w_{\text{global}}, Loss)$ to $DFactors$
12: **end for**
13: Sort $DFactors$ by $Loss$
14: **return** top $N$ discriminative factors

---

We include a formalized, abstract definition of DFactor in the final version to provide a clear conceptual paradigm beneficial to the research community. Formally, the definition of DFactor is as follows.

---

**Algorithm 2** Candidate Discriminative Factors Generation

**Input:** Time series data $\{h_k\}_{i=1}^{N}$, shapelet length $L$, number of candidates $M$
**Output:** Candidate shapelet set $\mathcal{C} = \{c_j\}_{j=1}^{M}$
1: Initialize an empty candidate set $\mathcal{C} \leftarrow \emptyset$
2: **for** each time series $h_k$ **do**
3:     a.     Extract all subsequences of length $L$: $SCO_i = \{sco_{i,k} \mid sco_{i,k} = [h_{i,k}, h_{i,k+1}, \ldots, h_{i,k+L-1}], \ k = 1, 2, \ldots, K_i\}$
4:     b.     Compute the mean subsequence: $s\bar{c}o_i = \frac{1}{K_i} \sum\limits_{k=1}^{K_i} sco_{i,k}$
5:     c. Initialize distance scores:
6:     **for** each subsequence $sco_{i,k} \in S_i$ **do**
7:         $D(sco_{i,k}) = \|sco_{i,k} - s\bar{c}o_i\|_2$
8:     **end for**
9: **end for**
10: Merge all subsequences and distance scores:
    $SCO = \bigcup\limits_{i=1}^{N} SCO_i, D(sco)$ for all $sco \in SCO$
11: **for** $j = 1$ to $M$ **do**
12:     a. Select the subsequence with the highest distance score:
        $sco^* = \arg \max\limits_{sco \in SCO} D(sco)$
13:     b. Add $sco^*$ to the candidate set: $\mathcal{C} \leftarrow \mathcal{C} \cup \{sco^*\}$
14:     c. Update distance scores:
15:     **for** all unselected subsequences $sco \in SCO$ **do**
16:         $D(sco) \leftarrow D(sco) + \|sco - sco^*\|_2$
17:     **end for**
18:     d. Mark $sco^*$ as selected: $D(sco^*) \leftarrow -\infty$
19: **end for**
20: **return** the candidate set $\mathcal{C}$

---

DFactor represents a feature vector derived from diffusion-based spatio-temporal decoupling, characterizing the variation patterns of specific categories. Specifically, DFactor partitions spatio-temporal information into K distinct classes based on feature similarity. Within each class, DFactor encodes variation patterns across superpixel regions during sampling. Consequently, these K classes of DFactors constitute a feature pattern library. For downstream classification tasks, relevant vectors obtained via identical spatio-temporal decoupling processes can be matched against this library to

achieve precise classification.

These principles can be formalized by the following equation:

$$\mathcal{L} = -g\left(S_1\left(\text{ DFactor }_1, \mathcal{T}_1\right), \ldots, S_K\left(\text{ DFactor }_K, \mathcal{T}_K\right)\right),$$
(17)

where $\mathcal{L}$ quantifies the dissimilarity among samples across K categories with respect to their DFactors. $S_i\left(\text{ DFactor }_i, \mathcal{T}_i\right)$ represents the set of distances related to a specific class $\mathcal{T}_i$ .The function $g(\cdot)$ takes K finite sets as input and outputs a scalar value indicating the overall dissimilarity among these sets.

## B. Example Analysis

To better illustrate the STD-FD method, we conduct an example analysis in this section. Specifically, we address three core questions related to the conceptualization, design, and implementation of STD-FD through detailed examples. These questions are: (1) *Q1: Why is temporal information introduced?* (2) *Q2: Why is spatial information extraction necessary*? (3) *Q3: Why use superpixels for segmentation*?

***Q1: Why is temporal information introduced?*** Through the truncation reconstruction process, we found that the reconstruction process is a dynamic changing process, and the number of steps and the amount of noise added have a significant impact on the reconstruction results. As shown in the Figure 6, when the reconstruction process is not truncated (i.e., with 20 time steps), it does not reflect the optimal point of the real and fake image reconstruction error. The optimal point may exist in the middle of the denoising process (as indicated by the pink box). Simply using the reconstruction results to judge the difference between positive and negative samples is incomplete. Therefore, we propose the introduction of temporal information.

***Q2&Q3: Why is spatial information introduced, and why use superpixels for segmentation***

We provide an intermediate process diagram for the denoising sampling process of normal and forged samples. Without the guidance of external prompts, the deterministic denoising process is essentially the process in which the diffusion model reconstructs the image through noise distribution. As shown in the figure 7, during the sampling process, noise is actually fitting the shapes and distributions of different semantic subjects. For example, some noise is used to construct the background information of the image, while other noise is used to construct the face and body. This leads us to consider two issues:

1.The degree of variation across different regions of the image is not the same. **If spatial information is not refined through segmentation, crucial information in small changes will be overlooked.** For example, in the figure above, the background of the real sample occupies a large proportion, but its variation is much smaller than that of facial features. Without block-based detailed modeling and analysis, the weight of facial changes in smaller regions would be low and might even be ignored.

2.**If the image is rigidly divided into blocks, the spatial correlation information will be lost.** For example, if we divide fake samples into blocks with fixed proportions, the background and the body of the woman will overlap. As seen in the image, the noise distribution changes differently for these two subjects. This is clearly inappropriate, so we adopted the superpixel segmentation algorithm, which creates internally self-correlated image blocks based on semantic information such as color, position, and contrast.

*Table 8.* Running Time and GPU Usage Comparisions.

| Detection Method | Running Time | GPU Usage |
|---|---|---|
| Xception | 250ms | 5010MiB |
| EfficientNet | 193ms | 3580MiB |
| F3Net | 274ms | 5100MiB |
| Conv-B | 267ms | 4995MiB |
| DIRE | 365ms | 4084MiB |
| STD-FD | 272ms | 2253MiB |

## C. Runtime and Resource Usage

We compare the runtime and memory usage of STD-FD with common baselines. Images are resized to 224×224 with batchsize of 32, and reconstruction-based methods used 20 denoising steps. Experiments are conducted on a Silver 4310 CPU and an NVIDIA A40 GPU. As shown in Table 8, even when considering the data reconstruction time, STD-FD's running time remains comparable to that of conventional network architectures. Furthermore, it demonstrates a significant advantage in terms of GPU usage, which can be attributed to the exceptional feature engineering employed in our method. Even with the use of simple classifiers such as logistic regression (LR), the model achieves outstanding performance results.

## D. Detailed Experimental Results

In this section, we provide the detailed experimental results referenced in the main text. Specifically, Table 9 and Table 10 present the results of STD-FD compared to all evaluation methods across all data subsets in the DeepFaceGen Benchmark and GenImage Benchmark experiments.

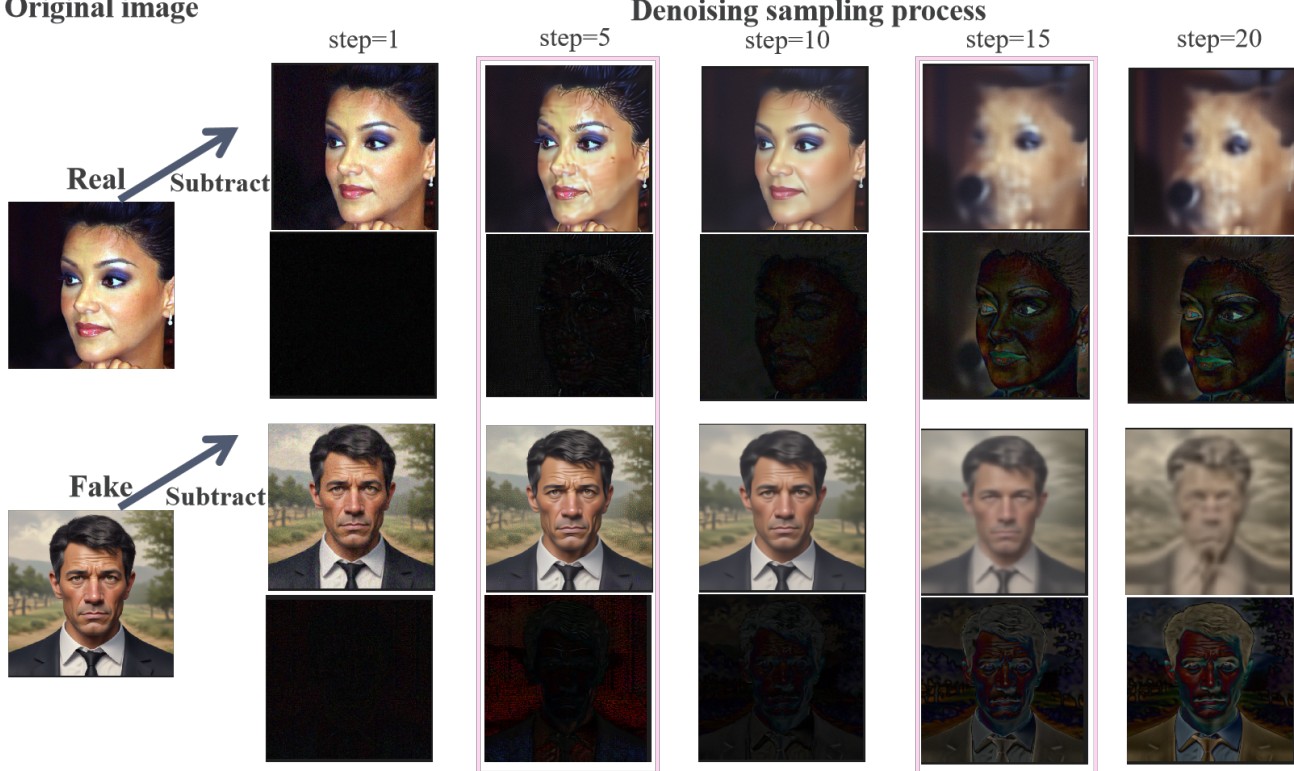

*Figure 6.* Truncated display of image reconstruction. It can be seen that the time step with the largest difference between the real and false samples does not exist in the result, but in the middle time step in the pink box.

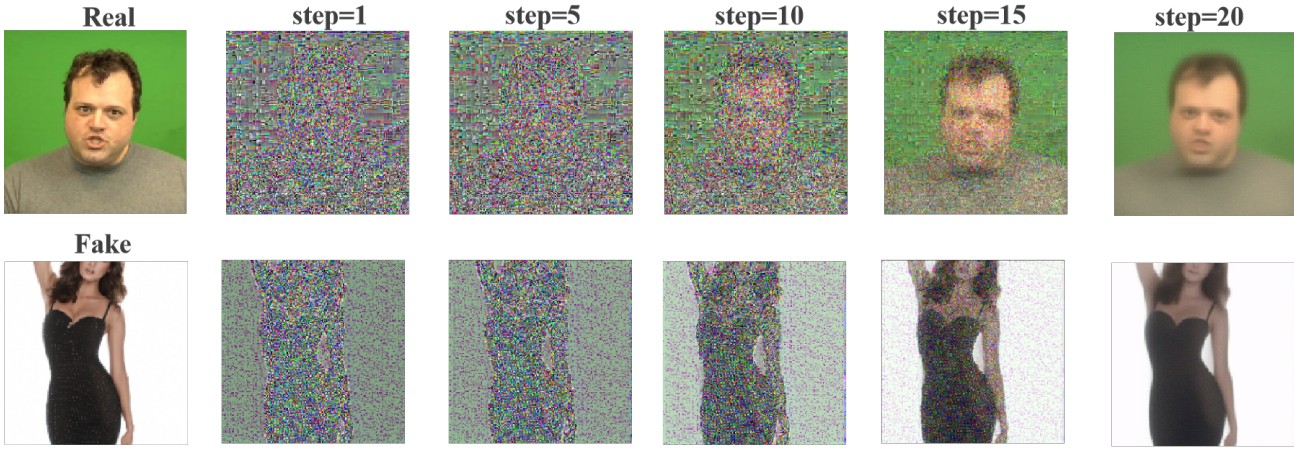

*Figure 7.* The visualization of noise map results during the image reconstruction process.

*Table 9.* Performance of Various Algorithms across Generators

| Generator | Xception | Efficientnet | F3net | RECCE | DNADet | Freqnet | DIRE | DCRT | UnivFD | NPR | STD-FD (ours) |
|-----------|----------|--------------|-------|-------|--------|---------|------|------|--------|-----|---------------|
| Midjourney | 77.01 | 79.52 | 81.65 | 87.64 | 93.44 | 85.69 | 87.01 | 89.78 | 88.67 | 89.01 | **94.36** |
| DALL·E 1 | 75.45 | 81.74 | 84.73 | 83.25 | 85.62 | 84.25 | 86.65 | 89.91 | 87.64 | 89.54 | **91.21** |
| DALL·E 3 | 86.59 | 88.41 | 87.23 | 89.17 | 83.90 | 87.13 | 87.84 | 88.05 | 89.21 | 89.41 | **90.01** |
| Wenxin | 86.87 | 84.34 | 89.28 | 92.13 | 91.60 | 91.98 | 92.84 | 92.72 | 90.01 | 92.35 | **92.90** |
| SD1 | 87.64 | 86.83 | 90.95 | 89.83 | 92.40 | 90.94 | 92.35 | 93.56 | 90.01 | 90.12 | **93.77** |
| SDXLR | 84.13 | 93.46 | 86.40 | 90.46 | 89.03 | 89.40 | 88.61 | 89.51 | 89.01 | 88.64 | **93.50** |
| OJ | 89.72 | 89.00 | 92.72 | 96.90 | 94.04 | 93.08 | 91.28 | 92.45 | 88.01 | 90.28 | **97.05** |
| pix2pix | 83.42 | 77.61 | 81.21 | 89.71 | 88.52 | 88.66 | 89.01 | 91.51 | 89.54 | 89.30 | **92.34** |
| SD2 | 87.79 | 85.91 | 89.11 | 95.43 | 92.70 | 90.92 | 89.54 | 90.41 | 91.45 | 90.01 | **96.71** |
| SDXL | 86.06 | 87.65 | 91.43 | 96.75 | 92.28 | 92.61 | 91.54 | 91.01 | 90.01 | 89.87 | **97.05** |
| VD | 85.02 | 83.84 | 89.52 | 95.67 | 89.21 | 96.55 | 98.78 | 94.25 | 89.68 | 91.01 | **100** |
| DF-GAN | 95.42 | 96.71 | 93.45 | 93.54 | 94.22 | 97.11 | 90.32 | 92.54 | 95.01 | 98.88 | **100** |
| Average | 85.42 | 86.25 | 88.14 | 91.70 | 90.58 | 90.69 | 90.69 | 90.48 | 91.30 | 89.85 | **94.90** |

*Table 10.* Comparison of Methods Across Various Forgery Generators Accuracy (ACC, % ) comparisons of our STD-FD and other generated image detectors. All methods were trained on GenImage/SDv1.4 and evaluated on different testing subsets.

| Method | Midjourney | SDv1.4 | SDv1.5 | ADM | GLIDE | Wukong | VQDM | BigGAN | Avg. |
|--------|-----------|--------|--------|-----|-------|--------|------|--------|------|
| CNNSpot | 84.92 | 99.88 | 99.76 | 53.48 | 53.80 | 99.68 | 55.50 | 49.93 | 74.62 |
| F3Net | 77.85 | 98.99 | 99.08 | 51.20 | 54.87 | 97.92 | 58.99 | 49.21 | 73.51 |
| CLIP/RN50 | 83.30 | 99.97 | 99.89 | 54.55 | 57.37 | 99.52 | 57.90 | 50.00 | 75.31 |
| GramNet | 73.68 | 98.85 | 98.79 | 51.52 | 55.38 | 95.38 | 55.15 | 49.41 | 72.27 |
| De-fake | 79.88 | 99.86 | 99.62 | 68.62 | 71.57 | 98.42 | 78.43 | 74.37 | 84.73 |
| Conv-B | 83.55 | **99.99** | 99.92 | 51.75 | 56.27 | 99.91 | 58.41 | 50.00 | 74.98 |
| Swin-T | 62.11 | **99.99** | 99.88 | 49.85 | 67.62 | 99.01 | 62.28 | 57.63 | 74.79 |
| UnivFD | 91.46 | 96.41 | 96.14 | 58.07 | 73.40 | 94.53 | 67.83 | 57.72 | 79.45 |
| DIRE | 50.40 | **99.99** | **99.92** | 52.32 | 67.23 | **99.98** | 50.10 | 49.99 | 71.24 |
| PatchCraft | 79.00 | 89.50 | 89.30 | 77.30 | 78.40 | 89.30 | 83.70 | 72.40 | 82.30 |
| AIDE | 79.38 | 99.74 | 99.76 | 78.54 | **91.82** | 98.65 | 80.26 | 66.89 | 86.88 |
| DRCT | 91.50 | 95.01 | 94.41 | 79.42 | 89.18 | 94.67 | 90.03 | 81.67 | 89.49 |
| STD-FD (ours) | **93.76** | **99.99** | 99.45 | **81.64** | 90.31 | 96.41 | **92.74** | **81.79** | **92.01** |

