# OpenReview forum: "STD-FD: Spatio-Temporal Distribution Fitting Deviation for AIGC Forgery Identification"
_ICML.cc/2025/Conference — ICML 2025 poster_

### Official Review · Reviewer_uUED · 2025-03-12

**Overall Recommendation:** 3

**Summary:**

In this paper, the authors propose a deepfake detection method based on “temporal distribution fitting deviations.” Specifically, they argue that existing reconstruction-based approaches treat the diffusion model as a black box, which limits their generalizability. In contrast, the authors decouple the sampling process and detect forgery features by predicting temporal inconsistencies across different time steps.

**Claims And Evidence:**

The claims in the methodology section are clear. However, the paper does not thoroughly discuss how the proposed method specifically addresses its motivation—for instance, how it effectively eliminates the strong coupling between the reconstruction model and the detection method. Additionally, it remains unclear whether the DFactor exists across all models and whether it follows the same distribution in different models.

**Essential References Not Discussed:**

N/A

**Experimental Designs Or Analyses:**

The experimental setting is sound.

**Methods And Evaluation Criteria:**

The evaluation metrics are feasible.

**Other Comments Or Suggestions:**

N/A

**Other Strengths And Weaknesses:**

In addition to the aforementioned unclear descriptions, there are additional concerns:

1. Regarding the “Mismatch Between Pre-trained Model and Identification Target”, while the authors demonstrate strong performance of the proposed method under this condition, it remains unclear why the method is able to mitigate this issue to such a significant extent. A more detailed explanation is needed to support this claim.

2. What specific diffusion model is used in STD-FD? Is this model interchangeable? If so, how does the detection performance change when a different diffusion model is used?

**Questions For Authors:**

If the authors can address the concerns, the reviewer would consider increasing the rating.

**Relation To Broader Scientific Literature:**

The proposed method should primarily be considered within the scope of Deepfake detection, especially for detecting forgeries generated by diffusion models. Whether it possesses strong detection capabilities and generalization ability for other types of forgeries remains an open question and requires further discussion.

**Theoretical Claims:**

The proposed pipeline, ie, DFactor set construction, DFactor selection, and forgery detection, is relatively clear. However, the reviewer is confused on the Spatial Information Capture. Specifically, why does the use of superpixels lead to improved performance in diffusion models? Is there any theoretical justification for this claim? The paper does not seem to provide a rigorous explanation, leaving this aspect unconvincing.

---

> ### Author Rebuttal · Authors · 2025-03-30
>
> Thank you for recognizing our work. Below are our responses to your questions (Q).
>
> **Q1: Methodological Coupling & DFactor Universality**
>
> Sorry for the misunderstanding. Reconstruction-based approaches rely on the magnitude of reconstruction error to distinguish real from fake images. When the reconstruction model encounters data from an unfamiliar domain, its robustness deteriorates. Our method instead uses a diffusion model to map images into a latent space, from which we **extract their temporal variations**.
>
> >To illustrate, consider a simplified scenario in which 5 and 80 represent pixel values from two distinct domains. Reconstruction-based methods attempt to reconstruct them (e.g., 5 → 4.9 and 80 → 70), leading to large errors when dealing with domain shifts (e.g., transitioning from cat to human, or from a diffusion-generated fake to a GAN-generated fake). In contrast, our approach **tracks the feature variation over time**, such as:
> 1 → 2 → 3 → 4 → 4.9 and 50 → 55 → 60 → 65 → 70. Although the numerical values differ greatly, both sequences share a **similar variation trend** (akin to a slope in mathematics or acceleration in physics). Based on this observation, we design a spatio-temporal distribution modeling framework centered on DFactor, thereby decoupling from semantic content.
>
> To clarify this principle, we visualized the average temporal changes of 256 pixels for cross-domain forgery data projected into latent space. The results show (Following the guidelines of this rebuttal, visualizations can be seen on anonymous website https://anonymous.4open.science/r/STDFD_re/README.md , if interested) :
>
> - Significant Differences between Real and Fake Data: Real images exhibit irregular, larger-scale changes due to the absence of fixed generative constraints, whereas forgeries conform to specific generative paradigms (GAN/diffusion/autoregressive) and have smaller, more uniform variation patterns.
>
> - Consistent Trends across Different Subjects and Architectures: Whether it’s a cat vs. a person, or a GAN vs. diffusion vs. autoregressive model, the amplitude of temporal change remains similar. **This amplitude (or trend) is precisely what motivates our design of DFactor**.
>
> Furthermore, the sampler we use was pretrained on general content, making it adept at lower-level semantic reconstruction. This capability facilitates an effective mapping of images into the latent space for our approach.
>
> **Q2: Superpixel Effectiveness**
>
> Qualitative: The superpixel algorithm segments images into semantically coherent regions based on similarities in color, texture, and pixel-level low-level features. For example, human figures and backgrounds naturally form distinct semantic regions. Superpixel-based segmentation effectively decouples their individual temporal variation patterns during sampling. A detailed analysis and illustration of this benefit can be found in Appendix B (see Figure 7).
>
> Quantitative: Experiments comparing superpixel and patch-based methods confirm improved performance. Superpixels outperform the patch-based and no-segmentation baselines by +2.04% and +3.15%.  Detailed experimental results on the 12 forgery subsets are provided on anonymous link ( https://anonymous.4open.science/r/STDFD_re/README.md , if interested).
>
> **Q3: Generalization Capability**
>
> Qualitative: Please refer to Q1.
>
> Quantitative: STD-FD achieves SOTA detection performance across two benchmarks involving GAN-based (DF-GAN, BGAN) and autoregressive-based models (DALLE series), supporting its general applicability.
>
>
> **Q4:  Mitigating Model-Target Mismatch**
>
> Please refer to Q1.
>
> **Q5: Diffusion Model Interchangeability**
>
> Thanks for this practical question. Due to computational efficiency and sampling quality, we primarily use DDIM. To validate robustness, we compared DDIM with alternative sampling methods (DDPM, DPM-Solver, and Progressive Distillation):
> | Method       | DALLE·1 | DALLE·3 | Midjourney | Wenxin | AUC Change | Sampling Time|
> |-------------|---------|---------|------------|--------|------------|-------------------|
> | DDPM| 86.4   | 86.3| 89.0| 87.3  | -5.8%| ~+43%|
> | DPM-Solver| 91.6| 90.7| 93.4| 95.6 | +0.3%| ~+4%|
> | Progressive Distillation| 89.6| 88.3 | 92.7| 93.9  | -1.5%| ~-27%|
> | DDIM (Baseline) | 91.4| 91.2| 94.0 | 93.4  | - | - |

---

### Official Review · Reviewer_918Q · 2025-03-12

**Overall Recommendation:** 4

**Summary:**

This paper proposes an AIGC forged image detection method based on Spatio-Temporal Distribution Fitting Deviation (STD-FD). For forged images, the authors decompose the spatio-temporal features of the generation process, employ superpixel segmentation to divide semantic units, and extract the DFactor in the spatio-temporal domain by combining the noise distribution changes of the diffusion model during the denoising process. Experiments show that STD-FD outperforms existing methods on well-known datasets such as GenImage and DeepFaceGen, especially in terms of cross-generator generalization and anti-post-processing robustness.

**Claims And Evidence:**

Yes

**Essential References Not Discussed:**

No.

**Ethical Review Flag:**

Flag this paper for an ethics review.

**Experimental Designs Or Analyses:**

Yes

**Methods And Evaluation Criteria:**

Yes

**Other Comments Or Suggestions:**

1. Line 166: "Noise Reshaping" should be "Noise Normalization."
2. Line 180: "GDTW" should include a citation, as should Line 271.
3. Line 273: $match(c_k, c)$ should be $match(a_k, c_k)$.

**Other Strengths And Weaknesses:**

**Strengths**: This work establishes a paradigm shift in AIGC detection through spatio-temporal noise distribution dynamics analysis during diffusion sampling—a marked departure from static artifact detection. The integration of superpixel-guided semantic unit analysis with distribution fitting deviations demonstrates methodological novelty. Superior robustness is evidenced by rigorous cross-domain validation (GenImage/DeepFaceGen) against diverse generators (SD, DALL-E), coupled with deployment-friendly efficiency (Xception-level inference speed). The well-structured presentation and open-sourced implementation ensure both conceptual clarity and technical reproducibility.


**Weaknesses**：While adversarial robustness is partially validated via FGSM/PGD attacks, the evaluation lacks coverage of emerging diffusion-specific adversarial perturbations (e.g., latent space manipulation attacks). Addressing targeted attacks against diffusion sampling mechanics would strengthen real-world applicability claims.

**Questions For Authors:**

1. How do the parameters of superpixel segmentation (e.g., the number of blocks $K$) affect performance?
2. What is the training overhead (GPU usage and training time) of the method?

**Relation To Broader Scientific Literature:**

Proactive defense mechanisms have targeted GAN/autoregressive architectures with notable success. The text-to-image revolution driven by diffusion models, however, renders conventional reconstruction-based detection (inherited from GAN-era paradigms) increasingly inadequate.

This work transcends the "reconstruction error"[1,2,3,4] doctrine by establishing spatio-temporal distribution discrepancy modeling through diffusion sampling dynamics—a strategic alignment with cutting-edge diffusion applications in image decomposition/editing[5,6]. The paradigm shift from artifact amplification to generative process deconstruction represents advancement in next-generation AIGC defense frameworks.

Reference:

[1]“DIRE for Diffusion-Generated Image Detection”(ICCV) (2023).

[2]"AEROBLADE: Training-Free Detection of Latent Diffusion Images Using Autoencoder Reconstruction Error"(CVPR) (2024).

[3]"DRCT: diffusion reconstruction contrastive training towards universal detection of diffusion generated images"ICML(2024).

[4]"Aligned Datasets Improve Detection of Latent Diffusion-Generated Images"ICLR(2025).

[5]"SwiftEdit: Lightning Fast Text-guided Image Editing via One-step Diffusion"CVPR(2025).

[6]"Preference Alignment on Diffusion Model: A Comprehensive Survey for Image Generation and Editing"arXiv preprint arXiv(2025).

**Theoretical Claims:**

Yes. The paper's core theoretical contributions are rigorously anchored in diffusion fundamentals (DDPM/DDIM frameworks), with Eq.(7) demonstrating proper alignment to established reverse process derivations.

The spatio-temporal modeling innovation (Eqs.12-14) introduces a mathematically sound mechanism for capturing temporal variation patterns via minimal-distance sampling. The discrepancy detection framework effectively operationalizes DFactors through three-dimensional feature engineering (matching, distance, correlation), forming a cohesive detection pipeline.

The theoretical scaffolding supports the empirical claims.

---

> ### Author Rebuttal · Authors · 2025-03-30
>
> We appreciate your recognition of our method's novelty and effectiveness. Below are our responses to your Questions (Q) and Weaknesses (W):
>
> **W: Addressing targeted attacks against diffusion sampling mechanics would strengthen real-world applicability claims.**
>
> Thank you for your insightful feedback. Following the same experimental settings described in the "Influence of Adversarial Attacks" section, we conducted additional experiments specifically focusing on adversarial perturbations during the diffusion sampling process. Concretely, adversarial noise (with L2-norm strengths of [0.01, 0.03, 0.05]) was injected at each timestep of the reverse diffusion process (20 steps in total). Experimental results are summarized below (AUC,%):
> | Perturbation | DALLE·1 | DALLE·3 | Midjourney | Wenxin | Performance |
> |-----------------------|---------|---------|------------|--------|--------------------|
> | 0.05                  | 88.01   | 89.98   | 93.84      | 92.57  | -2.5%              |
> | 0.03                  | 89.56   | 90.43   | 93.56      | 93.62  | -1.8%              |
> | 0.01                  | 91.67   | 91.42   | 92.90      | 90.42  | -2.0%              |
> | Original (Baseline)   | 91.45   | 91.20   | 94.01      | 93.48  | Baseline           |
>
> Under these adversarial conditions, the performance fluctuates within approximately 2.5%, demonstrating that the STD-FD identification mechanism remains robust against targeted sampling attacks.
>
> **Q1: How do the parameters of superpixel segmentation (e.g., the number of blocks K) affect performance?**
>
> Thank you for this constructive question. We performed an ablation study varying the number of superpixel blocks Kfrom the baseline setting K=10. The performance variation across different K values is within approximately 1.08%, as summarized below (AUC,%):
>
> | K  | DALLE·1 | DALLE·3 | Midjourney | Wenxin | Performance |
> |----|---------|---------|------------|--------|--------------------|
> | 1  | 90.46   | 90.11   | 92.89      | 92.67  | -1.08%             |
> | 5  | 91.45   | 90.87   | 93.76      | 94.01  | -0.01%             |
> | 10 (Baseline) | 91.45 | 91.20   | 94.01      | 93.48  | Baseline           |
> | 15 | 91.89   | 91.01   | 93.45      | 92.93  | -0.23%             |
> | 20 | 92.45   | 90.89   | 93.87      | 91.54  | -0.37%             |
>
> It's noteworthy that selecting a larger value of K does not always yield better results. Superpixel methods inherently suggest an optimal clustering number based on image content. In facial forgery scenario, the recommended K≈10 ensures effective semantic consistency; significantly deviating from this value negatively affects pixel-level semantic coherence and impairs spatio-temporal decoupling during diffusion sampling.
>
> **Q2: What is the training overhead (GPU usage and training time) of the method?**
>
> Thank you for your question. STD-FD involves diffusion sampling, DFactor construction, and downstream classification during training. The peak GPU memory usage during training is approximately 18.8GB. With a batch size of 32, the average training time per epoch is approximately 3 minutes 30 seconds (tested on NVIDIA A40 GPU with Intel Silver 4310 CPU).
>
> **Q3: Ethical Review**
>
> Sorry for the confusion. As stated in our Impact Statement, our research strictly targets deepfake detection, aiming to mitigate potential risks posed by generative AI technologies. We do not facilitate unethical use; rather, our work enhances reliability and security in detecting AI-generated content. Therefore, we respectfully submit that our research aligns with standard ethical guidelines.

---

> > ### Comment · Reviewer_918Q · 2025-04-05
> >
> > After going through the authors' rebuttal and the other reviewers' comments, I believe my previous concerns have been fully addressed —  especially with the new results on sampling perturbations, superpixel ablation studies, and practical applicability experiments. The novelty of this work remains significant, and its technical contributions are sufficiently demonstrated. I’m happy to reaffirm my recommendation to accept this paper.

---

> > > ### Author Response · Authors · 2025-04-06
> > >
> > > Thank you sincerely for your thoughtful evaluation and encouraging feedback regarding our work! We deeply appreciate the time and care you have invested in reviewing our manuscript. We will carefully revise the manuscript accordingly.

---

### Official Review · Reviewer_71VX · 2025-03-13

**Overall Recommendation:** 4

**Summary:**

This paper proposes STD-FD, a detection framework for AI-generated image forgeries that analyze spatio-temporal distribution deviations inherent in diffusion models' generative processes. By modeling how noise residuals evolve across temporal sampling steps and decomposing spatial patterns through superpixel segmentation, the method identifies discriminative features (DFactors) that reveal distribution mismatches between authentic and synthetic content. Unlike artifact-based approaches, STD-FD focuses on dynamic inconsistencies in the generation trajectory, achieving superior detection accuracy through systematic quantification of temporal noise propagation anomalies and localized spatial irregularities. The core innovation lies in bridging temporal modeling of diffusion behaviors with spatial forgery localization, offering a principled detection paradigm for evolving AIGC synthesis techniques.

**Claims And Evidence:**

Yes

**Essential References Not Discussed:**

No

**Experimental Designs Or Analyses:**

Yes

**Methods And Evaluation Criteria:**

Yes

**Other Comments Or Suggestions:**

1. Correct "In the recent years" to "In recent years."
2. Some sentences should be restructured for clarity, e.g., "By using spatio-temporal distribution fitting deviations..." could be clearer as "Spatio-temporal distribution fitting deviations capture changes in the generative process, enabling effective forgery detection."

**Other Strengths And Weaknesses:**

Strengths:
1.	Detection Paradigm Innovation: By leveraging a spatio-temporal diffusion framework, this method moves beyond prior reconstruction-error-based approaches (e.g., DRCT) to systematically decompose distribution discrepancies between real and forged images.
2. Implementation Framework: Rather than simply classifying noise-variant frames end-to-end, STD-FD builds a discriminative knowledge repository via joint spatio-temporal modeling. This effectively reframes forgery detection as a feature engineering task guided by repository comparisons.
3. Experimental Validation: Large-scale tests on DeepFaceGen and GenImage demonstrate strong performance. Ablation studies, tests with mismatched pre-trained models, and resistance to adversarial attacks underscore its robustness.
4. Deployment Feasibility: The open-source release ensures reproducibility. Its lightweight distribution-matching mechanism is resource-efficient, favoring real-world deployment.
Weaknesses:
1. Is the essence of the distribution fitting bias the temporal variation in noise distribution, or is it an implicit constraint of the generative model itself? Can the authors provide further clarification?
2. A formal or axiomatic definition of DFactor should be used to delineate its core principles, further separating conceptual essence from specific implementations.

**Questions For Authors:**

Please refer to the above comments.

**Relation To Broader Scientific Literature:**

The field of Deepfake detection has predominantly focused on amplifying artifacts through frequency/spatial domain transformations, while recent advances (LARE2 [CVPR'24], AEROBLADE [CVPR'24], DRCT [ICML'24]) explore image reconstruction-based paradigms.
This work innovates by addressing a critical oversight of existing end-to-end reconstruction approaches: their neglect of temporal dynamics during the reconstruction process. The key contribution lies in systematically modeling spatio-temporal distribution characteristics of genuine versus synthetic samples throughout Diffusion-based reconstruction trajectories, establishing more discriminative forgery signatures through principled analysis of generation process deviations.

**Theoretical Claims:**

Yes

---

> ### Author Rebuttal · Authors · 2025-03-30
>
> Thank you very much for your detailed review and constructive comments. We sincerely appreciate your recognition of the innovation and thoroughness of our work. Below are our responses to your Questions (Q) and Weaknesses (W):
>
> **Q1|W1: Is the essence of the distribution fitting bias the temporal variation in noise distribution, or is it an implicit constraint of the generative model itself? Can the authors provide further clarification?**
>
> We apologize for any confusion. The essence of the distribution fitting bias lies in the temporal variations of noise distributions. Although DDIM employs a diffusion-based architecture, we use intermediate DDIM timesteps solely to extract spatio-temporal distribution information inherent in real and forged data, rather than fitting specifically to a particular network architecture. Our experimental results provide strong evidence across two benchmarks, effectively detecting forgeries generated from different architectures, including DF-GAN and BigGAN (GAN-based) as well as the DALLE series (autoregressive-based).
>
> **Q2|W2: A formal or axiomatic definition of DFactor should be used to delineate its core principles, further separating conceptual essence from specific implementations.**
>
> Thank you for your valuable suggestion. We will include a formalized, abstract definition of DFactor in the final version to provide a clear conceptual paradigm beneficial to the research community.
>
> Formally, the definition of DFactor is as follows:
>
> > DFactor represents a feature vector derived from diffusion-based spatio-temporal decoupling, characterizing the variation patterns of specific categories. Specifically, DFactor partitions spatio-temporal information into **K** distinct classes based on feature similarity. Within each class, DFactor encodes variation patterns across superpixel regions during sampling. Consequently, these **K** classes of DFactors constitute a feature pattern library. For downstream classification tasks, relevant vectors obtained via identical spatio-temporal decoupling processes can be matched against this library to achieve precise classification.
>
> These principles can be formalized by the following equation:
> $
> \mathcal{L} = -g\left(S_1(\text{DFactor}_1, \mathcal{T}_1)\, S_2(\text{DFactor}_2, \mathcal{T}_2)\,\dots\,S_K(\text{DFactor}_K, \mathcal{T}_K)\right)
> $
> - $\mathcal{L}$ quantifies the dissimilarity among samples across **K** categories with respect to their DFactors.
> - $S_i(\text{DFactor}_i, \mathcal{T}_i)$ represents the set of distances related to a specific class $\mathcal{T}_i$.
> - The function $g(\cdot)$ takes **K** finite sets as input and outputs a scalar value indicating the overall dissimilarity among these sets.
>
>
> **Q3: Corrections of writing**
>
> Thank you for your careful reading and valuable feedback. We will carefully correct these writing issues in the final version.

---

### Official Review · Reviewer_9Lcb · 2025-03-14

**Overall Recommendation:** 2

**Summary:**

This work presents the Spatio-Temporal Distribution Fitting Deviation (STD-FD) method for detecting image forgery in AI-Generated Content (AIGC), specifically leveraging generative diffusion models. The authors designed DFactors, which capture deviations in temporal distribution during the diffusion process. Extensive experiments were conducted to analyze the effectiveness of the proposed method under various experimental settings.

**Claims And Evidence:**

The paper presents two core claims.

### 1. The proposed spatiotemporal distribution extraction framework requires further examination in terms of its validity and applicability.

#### 1.1 Validity of the Method

**Overly idealized assumptions about information acquisition:**
The method assumes access to update information at each time step of the diffusion process, specifically the noise term \( \epsilon \). However, in real-world scenarios—particularly in forgery detection—such information is extremely difficult to obtain. This assumption is overly strong, as it essentially grants direct access to the complete generative process of the model. If this information were available, one could infer the model’s sampling mechanism and potentially reconstruct the generative model itself, making additional forgery detection unnecessary.

#### 1.2 Potential Overdesign of the Method

The proposed approach involves multiple steps:
- Extracting variation information between diffusion time steps.
- Identifying specific patterns within these variations.
- Performing pattern matching to detect forged images.

This process is relatively complex and may contain unnecessary design redundancies. A simpler alternative might be more feasible. For instance, if intermediate results of the diffusion process were accessible, applying downsampling or other transformations directly on these results for classification might achieve similar or even superior performance. Thus, the paper should at least include such an approach as a baseline for comparative experiments.

### 2. Recommendations for Experimental Design

Although the paper conducts extensive experiments, additional ablation studies could provide a more comprehensive evaluation of the method’s effectiveness. These include, but are not limited to:
- **Comparing with simpler baseline methods:** For example, using intermediate diffusion model outputs for classification instead of employing a complex spatiotemporal distribution extraction framework.
- **Investigating robustness across different diffusion models, sampling strategies, and acceleration techniques:** This includes comparing distilled vs. non-distilled models to assess their impact on the method’s robustness.

**Essential References Not Discussed:**

N/A

**Experimental Designs Or Analyses:**

### **Issues in Experimental Analysis**

- The paper does not clearly define the specific task (e.g., **patch-level forgery detection** vs. **model attribution**). This lack of clarity makes it difficult to accurately assess the validity of the experimental design.
- As previously discussed, the method relies on an overly strong assumption—namely, access to intermediate diffusion time-step information. This assumption affects the interpretability of the experimental results.

### **Sufficiency of Experiments**

The paper conducts a large number of experiments, which is a notable strength. However, concerns remain regarding the validity of the experimental design, particularly in two key areas:

#### **Fairness of Experimental Design**
- The proposed method benefits from access to diffusion time-step information, which may provide it with an inherent advantage.
- In contrast, the baseline methods used for comparison do not have access to this information, potentially leading to an unfair comparison.
- The paper does not sufficiently discuss or address this unfairness in the experimental setup.

**Methods And Evaluation Criteria:**

### **Ambiguity in Task Definition**

A fundamental issue in the paper is the lack of a clearly defined task. In forgery detection, the forgery detection task can take multiple forms, such as:

- **Patch-Level Detection:** Determining whether a specific region of an image has been manipulated or forged.
- **Model Attribution:** Identifying which generative model was used to produce an entire image.

Since the paper does not explicitly specify which task the proposed method is designed for, it becomes difficult to assess the appropriateness of the method’s design and evaluation metrics. Furthermore, this ambiguity may lead to uncertainty in interpreting the experimental results, ultimately weakening the method’s applicability and generalizability.

### **Unjustified Methodological Design**

#### **Strong Assumption**
The method relies on access to intermediate time-step updates of the diffusion model, specifically \( \epsilon \). However, such information is typically unavailable in practical applications. As a result, the method is built on an overly idealized assumption, raising concerns about its feasibility in real-world forgery detection tasks.

#### **Questionable Necessity of the Method**
As previously mentioned, the proposed framework employs a complex spatiotemporal distribution extraction approach. However, more straightforward alternatives—such as directly classifying intermediate diffusion model outputs—may suffice. The paper does not provide comparative experiments to validate the relative advantages of its approach, casting doubt on the necessity of its methodological design.

### **Validity of Evaluation Metrics**

The work uses metric, such as classification AUC, which is a standard.

**Other Comments Or Suggestions:**

Typos in Eq. 15.

**Other Strengths And Weaknesses:**

### **Strengths**

1. **Large-Scale Experiments**
   - The paper conducts a substantial number of experiments, demonstrating thorough empirical validation.   This comprehensive experimental evaluation is a notable strength of the study.

---

### **Weaknesses**

1. **Unclear Task Definition**
   - The paper does not explicitly define the specific objective of the forgery detection task. For example:
     - Is the goal to detect whether a specific region of an image has been manipulated?
     - Or is it to identify which generative model produced the entire image?
   - The lack of task definition makes it difficult to evaluate the applicability of the proposed method.
   - The paper should clearly specify **which specific task the method is designed for** and **justify the choice of this task**.

2. **Strong Assumption on Accessibility of Intermediate Diffusion Steps**
   - The method assumes **access to intermediate time-step information of the diffusion process (e.g., \(\epsilon\) updates)**, which is typically **unavailable in real-world forgery detection scenarios**.
   - In practice, forgery detection usually relies on analyzing the final synthesized image without access to the generative process.
   - The paper should **justify the validity of this assumption** or explore the feasibility of the method **without this assumption**.

3. **Potential Overdesign of the Method**
   - The proposed framework involves **a complex spatiotemporal variation extraction pipeline**, including:
     - Computing similarity across different time steps.
     - Analyzing changes in similarity over time.
     - Identifying patterns in these changes.
     - Matching patterns with images using gradient descent.
   - This multi-step process introduces extra computational complexity and potential design redundancies.
   - **Lack of Exploration of Simpler Alternatives**:
     - If intermediate diffusion steps are accessible, why not directly use these frames with downsampling?
   - The paper should:
     - **Analyze the computational complexity of the method**.
     - **Compare the proposed approach with simpler alternatives** to demonstrate its necessity.

4. **Unclear Justification for Spatial Information Extraction Strategy**
   - The method extracts spatial information using a **superpixel-based approach** instead of a more conventional **patch-based** method.
   - The paper may clarify:
     - **Why was the superpixel approach chosen over simpler patch-based methods?**
     - **Has the effectiveness of different spatial extraction strategies been compared?**

5. **Uncertainty in the Method’s Generalizability Across Different Diffusion Sampling Strategies**
   - The method relies on intermediate updates within the diffusion process, but its effectiveness under different sampling strategies remains unclear.
   - **Potential Factors Affecting Performance**:
     - **Variation in sampling steps across models** (e.g., Model A samples in 10 steps, while Model B samples in 5).
     - **Differences in ODE-based solvers** (e.g., DDIM vs. DPM-Solver).
     - **Applicability to stochastic sampling methods** (e.g., the paper adopts a deterministic approach like DDIM—does the method work with stochastic sampling?).
     - **Effectiveness on distilled diffusion models**.
   - The paper does not explore these concerns and should:
     - **Conduct experiments to analyze the method’s robustness across different sampling strategies, step counts, and diffusion models**.
     - **Discuss the impact of various sampling techniques on the method’s effectiveness**.

**Questions For Authors:**

1. **What specific forgery detection task is the method designed for?**
   - Is the goal to detect local manipulations within an image or to attribute the image to a specific generative model?
   - What is the motivation behind choosing this particular task?

2. **How does the method remain feasible without access to intermediate diffusion steps?**
   - Given that real-world forgery detection scenarios do not typically provide access to diffusion process updates (e.g., \(\epsilon\)), how can the method be adapted to work without this assumption?  Is there empirical evidence supporting the validity of this assumption?

3. **Why is the proposed method designed with such a complex multi-step framework?**
   - What advantages does this pipeline offer over simpler alternatives (e.g., direct classification using intermediate diffusion outputs with downsampling)?  Has the computational complexity been analyzed to justify the need for each step?

4. **Why was a superpixel-based approach chosen for spatial information extraction?**
   - How does it compare to a more conventional patch-based method?

5. **How well does the method generalize across different diffusion sampling strategies?**
   - Does the method perform consistently when applied to diffusion models with varying step counts, solvers (e.g., DDIM vs. DPM-Solver), or stochastic sampling approaches?
   - Is the method still effective when used with distilled diffusion models?
   - Have experiments been conducted to assess the robustness of the approach under different sampling conditions?

**Relation To Broader Scientific Literature:**

The paper focuses on a highly specific topic—**forgery detection**. The study is related to broader fields such as **trustworthy AI** and **privacy protection**.

**Theoretical Claims:**

The work does not make any theoretical claims and, therefore, does not include theoretical analysis.

---

> ### Author Rebuttal · Authors · 2025-03-30
>
> Thank you for acknowledging our work. **However, there is a significant misunderstanding: we do not use intermediate steps from the forgery generation architecture (Model A). Instead, we employ a general diffusion model (Model B) to obtain its intermediate process. Without knowing the specific architecture used for generating fake images, we map the images into a latent space and model the spatio-temporal distribution based on that latent temporal information**. Below are responses to your questions (Q) and weaknesses (W):
>
> **Q1/W1: Clarification on Task Definition**
>
> Apologize for any ambiguity. Our work focuses on binary image authenticity detection by disentangling spatio-temporal distribution differences between real and synthetic images. This is achieved through analyzing time-step noise patterns via DDIM sampling (unrelated to forgery generation).
>
> **Q2/W2: Assumption on Intermediate Diffusion Information & Experimental Fairness**
>
> Sorry for the misunderstanding. We clarify that no assumptions are made regarding access to original generative models (Model A). Instead, our framework uses publicly available DDIM sampling (Model B) to capture temporal noise dynamics. As illustrated in Figure 1a, STD-FD integrates the sampling process natively, ensuring practical applicability. The acquisition of intermediate diffusion features constitutes our key innovation rather than an unfair experimental advantage.
>
> **Q3/W3: Alternative Methods & Complexity**
>
> Sorry for the confusion.
>
> a) Upon initially discovering the importance of temporal sampling differences, we indeed experimented with simpler alternatives. Initial experiments with 3D-Xception and ViT achieved 87.48% and 88.01% AUC on DeepFaceGen – comparable to SOTA but below our method. The key improvement is the careful decoupling of temporal discrepancies, inspiring the fine-grained design of our STD-FD approach centered around the DFactor module.
>
> b) STD-FD requires 272ms (2253MiB) per image (Line 761), comparable to Xception (253ms/2090MiB) and EfficientNet (241ms/1985MiB).
>
> **Q4/W4: Superpixel vs. Patch-based**
>
> Thanks for your question.
>
> Qualitative:  Superpixel segments images into regions with similar color, texture, and low-level features, producing blocks more consistent with pixel-level semantics than uniform grid partitioning. Such spatial processing is essential for effectively decoupling temporal noise maps. For example, superpixel segmentation can distinguish distinct temporal variation patterns between human subjects and backgrounds, as detailed in Appendix B (Figure 7 for a specific case analysis).
>
> Quantitative: Our ablation study demonstrates that superpixels outperform the patch-based and no-segmentation baselines by +2.04% and +3.15% AUC, respectively. Detailed experimental results on the 12 forgery subsets, following the guidelines of this rebuttal, are provided on an anonymous link (if interested: https://anonymous.4open.science/r/STDFD/README.md ).
>
> **Q5/W5: a) Experiments with different sampling conditions; b) Sampling steps; c) Sampling methods**
>
> Appreciate your advice.
>
> a) Yes, Influence of sampling steps (line 416) show improved detection performance with increasing timesteps (from T=5 to 50). Notably, even at T=5, the AUC surpassed 90%, outperforming SOTA. This underscores the efficacy of modeling forgery traces via spatio-temporal distributions.
>
> b) Please see a).
>
> c) We incorporated additional sampling methods, including DDPM, DPM-Solver, and Progressive Distillation. Experiments followed the identical setup used in the influence of sampling steps study (line 416):
> - Reverse process of DDPM involves a stochastic Markov chain with randomness at each step. With limited steps, noise errors accumulate significantly, reducing the quality and efficiency of spatio-temporal features compared to deterministic methods (ideal results require around 1000 steps).
> - DPM-Solver reformulates diffusion sampling as solving an ODE using higher-order solvers, providing excellent image reconstruction quality in just 20 steps. Although high-quality outputs and superior FID scores are achieved, computational overhead inevitably increases due to the higher-order ODE solution.
> - Progressive Distillation (PD) utilizes pretrained DDIM as teacher model, training a student model to mimic the teacher’s performance in fewer steps. Although sampling times significantly decrease, a performance drop of 1.5% is observed due to changing the optimization objective.
>
> | Method       | DALLE·1 | DALLE·3 | Midjourney | Wenxin | AUC Change | Sampling Time|
> |--------------|---------|---------|------------|--------|------------|---------------------|
> | DDPM| 86.4| 86.3| 89.0| 87.3| -5.8%| ~+43%|
> | DPM-Solver| 91.6| 90.7| 93.4| 95.6| +0.3% | ~+4%|
> | PD| 89.6| 88.3| 92.7| 93.9| -1.5%| ~-27%|
> | DDIM (Baseline) | 91.4| 91.2| 94.0| 93.4  | -  | - |
>
> **W6: Equation Corrections**
>
> Thanks for meticulous review. We will rectify all notation inconsistencies in the final version.

---

### Decision · Program_Chairs · 2025-05-01

**Decision:**

Accept (poster)

**Comment:**

The submission received one weak reject, two accepts, and one weak accept. The authors provided detailed responses, which, while not overturning the negative rating, convinced the Area Chair that the submission's strengths and contributions outweigh its weaknesses.